# Fingolimod Prevents Neuroinflammation but Has a Limited Effect on the Development of Ataxia in a Mouse Model for SCA1

**DOI:** 10.3390/ijms26104698

**Published:** 2025-05-14

**Authors:** Chen Yang, Nienke Gravendeel, Amy Chin On, Laura Post, Ryan van Bergen, Catarina Osorio, Martijn Schonewille

**Affiliations:** Department of Neuroscience, Erasmus Medical Center, 3015 GE Rotterdam, The Netherlands; c.yang.2@erasmusmc.nl (C.Y.); nienkegravendeel@gmail.com (N.G.); amy.chin.on@student.uva.nl (A.C.O.); l.post@erasmusmc.nl (L.P.); r.vanbergen.1@erasmusmc.nl (R.v.B.); c.osorio@erasmusmc.nl (C.O.)

**Keywords:** spinocerebellar ataxia type 1, neuroinflammation, fingolimod, microglia, sphingosine-1-phosphate pathway

## Abstract

Spinocerebellar ataxia type 1 (SCA1) is a neurodegenerative disorder that predominantly affects the Purkinje cells (PCs) of the cerebellum, leading to cerebellar degeneration, motor dysfunction, and cognitive impairment. Sphingosine-1-phosphate (S1P) signaling, known to modulate neuroinflammation, has been identified as a potential therapeutic target in SCA1. To investigate the therapeutic efficacy of the S1P modulator fingolimod, we treated a mouse model for SCA1, ATXN1[82Q]/+ mice during three different periods with fingolimod and assessed the effects. Potential therapeutic effects were monitored by tracking locomotion during the treatment period and examining PC morphology, connectivity, and markers for neuroinflammation post-mortem. Fingolimod treatment reduced astrocyte and microglial activation during all three treatment periods. We found no effect on calbindin levels or the thickness of the molecular layer, but fingolimod did improve the extent of the synaptic input of climbing fibers to PCs. While fingolimod improved important aspects of cellular pathology, we could only detect signs of improvement in the locomotion phenotype when treatment started at a later stage of the disease. In conclusion, fingolimod is able to mitigate neuroinflammation, preserve aspects of PC function in SCA1, and remediate part of the ataxia phenotype when treatment is appropriately timed. Although behavioral benefits were limited, targeting S1P pathways represents a potential therapeutic strategy for SCA1. Further studies are needed to optimize treatment regimens and assess long-term outcomes.

## 1. Introduction

Spinocerebellar ataxia type 1 (SCA1) is a neurodegenerative disease caused by a mutation in the *Atxn1* gene and characterized by progressive ataxia, spasticity, ophthalmoplegia, and bulbar symptoms. In addition to these motor symptoms, patients often experience sensory symptoms and cognitive impairments in the later stages of the disease [1]. These symptoms profoundly impact the patient’s quality of life. The global prevalence of spinocerebellar ataxias is estimated to be between 1 and 5 per 100,000 individuals, while in Europe, the prevalence ranges from 0.9 to 3 per 100,000, with some regional variations [2]. The pathophysiological characteristics of SCA1 primarily include the degeneration of the cerebellar cortex, deep cerebellar nuclei, and brainstem [3,4]. At the cellular level, notable findings include myelin loss and the early activation of astrocytes and microglia. On a molecular level, SCA1 is attributed to an abnormal expansion of the CAG repeat sequences within the coding region of the *Atxn1* gene. In unaffected individuals, this region typically comprises 6 to 39 CAG repeats and 1 to 3 CAT interruptions, which are believed to contribute to the stability of the trinucleotide sequence during DNA replication [5]. SCA1 patients have 40 to 83 CAG repeats without interruptions, coding for polyglutamine (polyQ). The greater the number of repetitions, the earlier the onset and the shorter the patient’s life expectancy [6]. PolyQ primarily affects Purkinje cells (PCs) in the cerebellum, leading to their dysfunction and degeneration, contributing to balance issues and ataxia [7,8,9,10]. Different mouse models mimic different pathogenic mechanisms and focus on disease symptoms. For example, the motor defects in the transgenic SCA1 mouse line ATXN1[82Q], which selectively overexpresses the mutant ATXN1[82Q] protein in cerebellar PCs, provide support for the dominant role of cerebellar dysfunction in ataxia [11]. In addition, the symptoms in two SCA1 knock-in mouse lines (ATXN1[154Q/2Q] and ATXN1[78Q/2Q]) are related to the number of poly-Q repeats, like the symptoms of SCA1 patients. For example, while ATXN1[154Q/2Q] knock-in mice exhibit apparent motor deficits, such as impaired rotarod performance as early as 12 weeks, ATXN1[78Q/2Q] mice remain largely asymptomatic, which mirrors the clinical observation in SCA1 patients that longer polyglutamine tracts lead to an earlier onset and more severe disease phenotype [12]. Here, because we focused on the cerebellum-behavior changes in mice, like motor coordination and balance, we used the ATXN1[82Q] SCA1 mouse model, which expressed the mutated *Atxn1* only in PCs. This is a mouse model for SCA1 that permits precise investigation of cell-autonomous effects of the ATXN1 risk variant within Purkinje neurons. However, it does not recapitulate the ubiquitous expression of the variant observed in human carriers, thus limiting the extrapolation of findings to the systemic context of the human disease. In this mouse model, movement impairments begin to develop after 5 weeks of development [13,14,15,16]. Around 2 to 3 weeks, PCs begin showing reduced intrinsic excitability along with subtle dendritic atrophy and weakened metabotropic glutamate receptor 1 (mGluR1) signaling; these early functional impairments precede overt motor deficits and later structural degeneration in this mouse model [14,17]. We also found synaptic connections of climbing fibers were impaired before apparent ataxia developed [17,18]. Although several potential therapeutic approaches have been tested [14,19,20,21,22], there is currently no accepted treatment option for SCA1. There is great potential in compounds that inhibit toxic gain-of-function mechanisms associated with polyglutamine expansion, promote protein degradation pathways, or prevent mutant protein aggregation [23,24]. For example, antisense oligonucleotides (ASOs) that inhibit the expression of the mutant *Atxn1* gene have been developed, but phase III trials still have to be completed [6]. An alternative or complementary path is to reduce neuroinflammation by regulating microglial activation [25,26,27]. This path has proven its potential to prevent neuronal loss in preclinical models and clinical settings [28,29,30].

We have recently shown that sphingolipid-1-P expression is specifically increased in the anterior mouse cerebellum [31]. Sphingosine-1-phosphate (S1P) and its receptors (S1PR) are bioactive lipid molecules that are widely expressed, including in the human body, and play a crucial role in regulating the immune response [32]. S1P is a signaling sphingolipid, produced by the enzyme sphingosine kinase 1 (SPHK1) and released by neurons throughout the central nervous system (CNS). Its receptors are expressed by all types of brain cells, including microglia, astrocytes, and neurons [33,34]. The discovery of the presence of the S1P/S1PR complex in the CNS has led to the successful exploitation of signaling pathways that modulate this complex as a therapeutic approach for neurological diseases. We found evidence for a potential role in SCA1, when we deleted the *Sphk1* gene in ATXN1[82Q]/+ mice in previous work and observed that SPHK1 deficiency affected ATXN1[82Q] protein expression and prevented PC degeneration [31]. Expression of both astrocytes and microglia in the cerebellum was reduced, resulting in reduced inflammatory responses in the mouse brain [31]. These findings suggest that inhibiting the S1P pathway and the associated reduction of inflammatory reactions could be a potential therapeutic approach. To apply this finding to the treatment of SCA1, we decided to treat mice with fingolimod.

Fingolimod, previously known as FTY720, is an immunomodulatory drug primarily acting as a functional antagonist of sphingosine-1-phosphate (S1P) receptors, and is approved for the treatment of neuroinflammatory diseases [35]. Based on its lipophilic nature, fingolimod crosses the blood-brain barrier, where the drug inhibits S1P receptors on neuronal cells, particularly astrocytes, to reduce astrogliosis [36]. Fingolimod’s primary mode of action involves phosphorylation by sphingosine kinases and subsequent interaction with S1P receptors; the precise mechanisms by which fingolimod exerts its neuroprotective effects remain incompletely understood. Currently, fingolimod is used to treat multiple sclerosis (MS) [37]. Taken together, this suggests fingolimod treatment may reduce the immune response in SCA1 mice, either by affecting PCs and/or via modulation of glial responses by affecting the S1P pathway.

Here, we tested the hypothesis that fingolimod treatment can alleviate motor dysfunction and inflammatory responses in a SCA1 mouse model by inhibiting the S1P pathway. Based on previous results [13,14,17], we tested different treatment regimens, starting either at the onset of the cell activity phenotype (week 2, 8 weeks treatment, regimen 1), at the onset of the behavioral phenotype (week 5, 8 weeks treatment, regimen 2) or at the later stage of the disease (week 10, 4 weeks treatment, regimen 3). Based on the results obtained from the experiments involving these three different treatment regimens, we concluded that fingolimod can reduce immune and inflammatory responses in SCA1 mice and improve PC connectivity. The behavioral improvements in motor function required a later onset of treatment and were limited to part of the ataxia phenotype.

## 2. Results

### 2.1. Minimal Improvement in Behavioral Deficits of SCA1 Mice Following Fingolimod Treatment

Based on the beneficial effect of deleting *Sphk1* in ATXN1[82Q] mice, we aimed to test the therapeutic potential of the S1P-receptor modulator fingolimod in these mice [31]. In ATXN1[82Q] mice, impaired balance beam or rotarod performance deficits are typically not observed until 5–6 weeks after birth [14,38]. However, previous work indicates that cellular activity and plasticity are affected earlier, as early as 2 weeks after birth [17]. Hence, to investigate whether fingolimod delays or prevents motor impairment in SCA1 mice at that early stage of the disease, we administered intraperitoneal injections at 2 weeks of age, three times a week, for 8 weeks. Prolonged treatment periods with fingolimod are possible, as currently, fingolimod treatment in MS patients can be for durations up to years [37]. From 4 weeks of age, rotarod tests and two different diameter balance rod tests were performed once a week to minimize overtraining (Figure 1A). Mice were handled and trained for 1 day before the experiment to help them become familiar with the experimental setup and paradigm. In the rotarod test, we observed a behavioral deficit in SCA1 mice starting from the 6th week. We did not find noticeable behavioral improvements in the rotarod test due to the fingolimod treatment (Figure 1B,C). The 12 mm balance beam test showed a behavioral deficit between control and SCA1 mice at 7 weeks. Throughout the testing period of the 12 mm balance beam experiment, we did not find evidence of a positive effect of fingolimod on the behavioral impairments of SCA1 mice (Figure 1D). The number of foot slips in SCA1 mice was significantly higher than that of control mice, without a significant effect of fingolimod treatment on SCA1 mice (Figure 1E). Finally, we tested the mice on the 6 mm balance beam. In this more difficult task, SCA1 mice already showed deficits in balance and coordination by the 5th week. There was also no effect of fingolimod treatment, except for a single, significant improvement in the final 10th week; fingolimod treatment improved motor performance in SCA1 mice (Figure 1F). Finally, in the analysis of the foot slips, we found that fingolimod significantly reduced the number of foot slips in SCA1 mice at week 10 (Figure 1G). Our data show that SCA1 mice have behavioral impairments from week 5, which aligns with previous work. Fingolimod treatment did not improve motor behavior in the first weeks of treatment but alleviated behavioral impairments in SCA1 mice at the later stages of our analysis.

### 2.2. Fingolimod Attenuates the Inflammatory Responses in ATXN1[82Q] Mice

To evaluate the effects of fingolimod treatment, we perfused the mice and obtained cerebellar tissue for immunofluorescence directly after completing the behavioral tests at week 10. Because disease progression in the SCA1 mouse model is characterized by changes in astrocytes and microglia [4,26,39], and fingolimod is a competitive inhibitor of the S1P pathway and mainly affects the immune response, previous studies have tested and shown that deletion of sphk1 results in a reduced number of microglia. We used GFAP to label astrocytes and Iba1 to label microglia and evaluate possible differences [4,40,41]. We quantified the fluorescence intensity using the intensity and variance of the intensity. The intensity was taken as a measure for the amount of protein, whereas the SD of the intensity indicates the variance in protein expression across a sample or region of interest. We found that the variance in GFAP fluorescence expression, measured by the standard deviation of the intensity, is elevated in SCA1 mice compared to control mice at 10 weeks. Treating mice with fingolimod reduced the variance in GFAP expression in SCA1 mice (Figure 2E), an effect observed in both the anterior and posterior cerebellum (Appendix A). In line with the GFAP/astrocyte analysis, Iba1 intensity was quantified across both the anterior and posterior cerebellum, whereas microglia cell counts were performed specifically in anterior cerebellum, based on our previous findings in this model [31]. This region-specific approach enabled us to capture both more general and region-specific changes. The intensity and variance of the intensity of Iba1 were similar or even reduced in SCA1 mice (see Figure 2F and Appendix A). Fingolimod treatment had a dichotomous effect on Iba1 intensity, increasing Iba1 intensity levels in control mice and decreasing them in SCA1 mice after treatment (Figure 2F). Visual inspection suggested that, rather than intensity parameters, the number of microglia was affected more by the mutation, as previously observed. The apparent contradiction between the intensity and number of microglia could be caused by the fact that the changes in intensity occur later and are a more indirect measurement, whereas the number of microglia more directly reflects changes in glia. The effect of deletion of *sphk1* was most pronounced in the anterior cerebellum [31], so we counted the number of microglia in the anterior cerebellum. SCA1 mice have, on average, ~35% more microglia than control mice (21.16 ± 5.82 vs. 33.37 ± 4.19, *p* < 0.001). In contrast to the behavioral deficits, treating mice with fingolimod significantly decreased the number of microglia in SCA1 mice (Figure 2G).

Next, we immunostained sections with Calbindin to label PCs and the molecular layer, and VGLUT2 to label the climbing fiber onto PCs to assess changes in morphology and connectivity (Figure 3A–D). Calbindin is a PC-specific marker, the expression levels of which slowly reduce with age in SCA1 mice starting from 6 weeks [11]. VGLUT2 labels the input from the inferior olive via the climbing fibers to PCs, a connection that consists of hundreds of synapses that span most of the PC dendrite. We found that the intensity of Calbindin staining and the thickness of the molecular layer were reduced in SCA1 mice, and these deficits could not be remediated by treatment with fingolimod (Figure 3E,F). The percentage of the PC dendrite in the molecular layer innervated by the climbing fiber (CF/ML%), indicated by VGLUT2 staining, is also reduced in SCA1 mice. Climbing fiber input is crucial for learning, and reduced climbing fiber innervation is linked to learning deficits. Interestingly, this pathological feature was remediated by fingolimod treatment in SCA1 mice (Figure 3G).

In summary, we found that fingolimod reduced the inflammatory response in the cerebellum of SCA1 mice mainly by decreasing the response of astrocytes, measured by the variance in GFAP staining. In addition, fingolimod also increases the area of PC dendrites contacted by climbing fibers.

### 2.3. Delaying the Start of Treatment with Fingolimod Has a Minimal Positive Effect on the Behavioral Impairment in SCA1 Mice

Next, we delayed the onset of the treatment period based on the previous dataset’s positive effect at week 10. As a result, the start of the treatment was aligned with the onset of the locomotion phenotype at 5 weeks, while again treating mice for 8 weeks (Figure 4A). Mice were handled and trained for one day before the start of behavioral testing to ensure acclimation to the rotarod and balance beam. Here, the rotarod behavioral deficits in SCA1 mice became evident by the 5th week. Similar to the previous treatment scheme, no significant improvements in rotarod performance were observed following fingolimod treatment (Figure 4B). In the 12 mm balance beam test, we first measured the time required for mice to cross the beam. The behavioral deficit between control and SCA1 mice emerged during the 8th week of testing. Fingolimod treatment did not yield significant improvements in these deficits (Figure 4C). Similarly, when analyzing foot slips on the 12 mm beam, SCA1 mice consistently exhibited more slips than control mice. Interestingly, fingolimod treatment exacerbated behavioral deficits in weeks 8 and 9, suggesting that early use of fingolimod to suppress inflammatory responses may have adverse effects earlier in disease progression. However, by the 12th week, fingolimod significantly reduced foot slips in SCA1 mice (Figure 4D). Finally, performance was assessed using the 6 mm balance beam test (Figure 4E), where SCA1 mice displayed behavioral deficits as early as week 8. Neither the time-to-cross analysis nor the foot slips results provided strong evidence for fingolimod’s efficacy in reducing motor impairments in SCA1 mice for this task, except for a small, but significant effect observed in the 12th week. Interestingly, we found that in the early disease stage, fingolimod increased the number of foot slips in SCA1 mice (Figure 4F).

In summary, our findings suggest that earlier in disease development, fingolimod treatment may amplify behavioral impairments in SCA1 mice. In contrast, fingolimod appeared to alleviate behavioral deficits when administered during the later stages of the disease.

### 2.4. Long-Term Use of Fingolimod Primarily Decreases Astrocyte Expression in SCA1 Mice in the Mid to Late Stage of the Disease

We perfused the mice and obtained cerebellar tissue for immunofluorescence after the behavioral test at week 13 to evaluate the effects of fingolimod treatment. We used GFAP to label astrocytes and Iba1 to label microglia (Figure 5A–D). Our analysis showed that the variance in GFAP fluorescence expression, measured by the standard deviation of the intensity, was increased in SCA1 mice compared to control mice at 13 weeks. However, fingolimod treatment only reduced the variance of GFAP expression in the anterior cerebellum of SCA1 mice, not in the posterior cerebellum or overall (Figure 5E and Appendix A). In contrast, analysis of GFAP fluorescence intensity showed that fingolimod reduced GFAP expression in SCA1 mice (see Appendix A). Next, we measured the intensity of Iba1 fluorescence and found that, unlike in younger mice (Figure 2), fingolimod treatment altered microglial Iba1 expression and standard deviation of expression only in the posterior cerebellum. Fingolimod treatment also resulted in a decrease in the number of microglia in both control and SCA1 mice (Figure 5F,G). Finally, we used Calbindin and VGLUT2 to examine PC health, molecular layer thickness, and the coverage of PC dendrites with climbing fiber synapses (Figure 6A–D). Long-term fingolimod treatment from 5 to 13 weeks of age did not increase calbindin staining intensity or the height of the molecular layer. Still, it did increase the area of the molecular layer covered by climbing fiber input in SCA1 mice (Figure 6E–G).

In conclusion, fingolimod treatment alleviated the inflammatory response in the cerebellum of SCA1 mice by inhibiting GFAP expression and the number of microglia. In addition, fingolimod also improved the input of climbing fibers but could not restore calbindin expression or the molecular layer in SCA1 mice at 13 weeks.

### 2.5. Short-Term Administration of Fingolimod Decreases Behavioral Deficits in SCA1 Mice in the Late Stages of the Disease

The previous experiment demonstrated that the administration of fingolimod can dramatically suppress the inflammatory response in SCA1 mice but can exacerbate behavioral deficits early in disease development. In the third and last experiment, we aimed to avoid potential negative effects of early fingolimod administration by treating mice only in the second half of the eight-week period. Thus, we performed intraperitoneal injections with fingolimod and vehicle in control and SCA1 mice, starting from 10 weeks of age, three times a week for 4 weeks, while conducting behavioral experiments (Figure 7A). In the rotarod test, SCA1 mice showed apparent behavioral deficits in week 11. No difference was observed at the ten-week time point, presumably because this time point represents the first exposure of the mice to the rotarod. Our experiment design employs a longitudinal rotarod paradigm with short sessions consisting of three runs across multiple timepoints, in contrast to protocols that assess performance on a single time point, over multiple runs on consecutive days to establish learning curves. Here, too, no significant behavioral improvement was observed in fingolimod-treated mice using the rotarod (Figure 7B). In the 12 mm balance beam test, SCA1 mice did show behavioral deficits at 10 weeks of age. Still, fingolimod did not significantly improve the motor deficits of SCA1 mice (Figure 7C). The SCA1 mice also had more foot slips than control mice, but fingolimod treatment did not affect that either (Figure 7D). In the 6 mm balance beam test, the behavioral deficits between the control and SCA1 groups were again significant. Interestingly, fingolimod treatment successfully improved the number of foot slips at the eleven-week time point and the latency to cross in all time points in SCA1 mice (Figure 7E,F).

In summary, our results suggest that short-term use of fingolimod during later stages of the disease can improve specific aspects of behavioral deficits in SCA1 mice.

### 2.6. Short-Term Treatment with Fingolimod Is Also Capable of Reducing the Inflammatory Responses in SCA1 Mice

After completing the behavioral testing at week 13, mice were perfused, and the cerebellum was examined using immunofluorescence. We found that the variation in GFAP fluorescence expression in the cerebellum of SCA1 mice was higher than that in control mice (Figure 8E). Fingolimod reduces the variance in the expression of GFAP in the cerebellum of SCA1 mice. Similarly, the expression of Iba1 in the cerebellum of SCA1 mice was significantly reduced after fingolimod treatment (Figure 8E and Appendix A). Counting the number of microglia in the anterior cerebellum confirmed the beneficial effect of fingolimod treatment in SCA1 mice, which decreased the number of microglia compared to mice injected with vehicle (Figure 8F,G). Finally, using calbindin and VGLUT2 staining, we examined PCs and climbing fiber synapses. We found that short-term use of fingolimod did not remediate the decrease in calbindin staining intensity or molecular layer thickness but was able to increase the molecular layer covered by climbing fiber input (Figure 9E–G).

In summary, we found that fingolimod mainly alleviated the inflammatory response in the cerebellum of SCA1 mice, measured by reduced variance in GFAP staining, the intensity of Iba1, and the number of microglia. In addition, fingolimod can also improve the input of climbing fibers but does not affect the thickness of the molecular layer in SCA1 mice.

## 3. Discussion

In this study, based on the positive effects of deleting sphingosine kinase 1 (*Sphk1*) in the ATXN1[82Q] SCA1 mouse model, we examined the potential beneficial effects of fingolimod treatment on disease progression in an ATXN1[82Q] SCA1 mouse model. In mutant mice, initial locomotor deficits and subtle cerebellar immune responses emerge around 6–8 weeks, progressively intensifying to culminate in significant morphological alterations in the molecular layer by 12–16 weeks [13,14,17]. We treated mice in three experiments with fingolimod for 4 to 8 weeks with three weekly injections at a level of 1 mg/kg body weight [38]. These regimens were designed to target different disease stages: an early stage before significant PC loss (2–10 weeks), an intermediate stage where mild behavioral deficits were evident (5–13 weeks), and a later stage corresponding to established degeneration (10–13 weeks). We aimed to evaluate the timing of anti-inflammatory interventions to influence treatment efficacy.

Throughout all three experiments, fingolimod consistently decreased the immune response, typically visualized by reducing the number of microglia and the variance in astrocyte GFAP staining. Fingolimod also improved the input of climbing fibers, suggesting a restoration of synaptic function. However, its inability to alter molecular layer thickness or PC calbindin levels indicates that the drug does not reverse the underlying structural degeneration of the cerebellar circuitry. Despite minor improvements, fingolimod did not consistently or substantially improve the locomotion phenotype in SCA1 mice, tracked using the rotarod and balance beam. More specifically, long-term treatment with fingolimod starting from the early stages of the disease was associated with an increase in behavioral deficits, whereas shorter treatment with fingolimod starting at a later stage significantly alleviated part of the locomotion impairment. One potential explanation for this difference could be that early in the disease, inflammation may serve an adaptive or compensatory role, and its suppression could interfere with protective astrocyte responses [4,42]. In contrast, at later stages, neuroinflammation then potentially shifts towards a maladaptive, chronic response characterized by gliosis and neurotoxicity, in which fingolimod’s modulation of S1P signaling may help restore homeostasis and limit damage. Interestingly, this stage-dependent shift in inflammatory function has been proposed before, in SCA1 treatment [4], and thus putatively underlies the differential effects observed across treatment windows.

Fingolimod is a sphingosine-1-phosphate (S1P) receptor modulator. S1P, a signaling sphingolipid, and its pathway play an important role in inflammation, with S1P being produced by SphKs and acting through specific G protein-coupled receptors (S1PRs) [43]. In microglia, S1P signaling has been shown to enhance the production of proinflammatory cytokines, such as TNF-α and IL-1β, leading to an enhanced inflammatory response [44]. Inhibition of *SphK1* results in decreased levels of these cytokines [45,46], suggesting that the S1P pathway is critical for mediating microglial activation during inflammation. S1P receptor 2 (S1PR2) has been identified as a mediator of microglial activation, implicated in proinflammatory responses [47]. Moreover, astrocytes express S1PRs and thus can also respond to S1P signals [48].

In the current study, fingolimod treatment consistently affected astrocytes and microglia, although the effects differed between both glial subtypes. In astrocytes, fingolimod decreased the variability of the GFAP signal following therapy after 2 to 10 and 10 to 13 weeks treatment and the GFAP signal in all three treatment regimes. Although microglia typically display an increased level of Iba1 (except for the first experiment), the most consistent phenotype of SCA1 disease development was an increase in the number of microglia. We previously demonstrated that genetic deletion of *Sphk1* can prevent those changes in astrocytes and microglia [31]. Treatment with fingolimod replicated that therapeutic effect, although the deficits were not always completely reversed as with the genetic deletion. Extending that previous finding, we also examined the effects on morphological features and connectivity, impacting SCA1 patients [49] and mouse models [17]. Fingolimod treatment was able to reverse the deficit in climbing fiber input to near-normal levels but did not affect calbindin levels or the thickness of the molecular layer, as the genetic deletion of Sphk1 was able to achieve previously. Interestingly, in the cerebellum, Bergmann glia are a prominent type of astrocyte that strongly impact the climbing innervation of PCs [50]. If Bergmann glia control both the immune response and the climbing innervation, this could be taken as evidence that fingolimod has less to no impact on PCs but primarily affects glial cells.

Finally, we performed a longitudinal analysis of the development of behavioral deficits in SCA1 mice treated with fingolimod, looking for potential improvements in cerebellum-dependent behavior like motor coordination and balance. We found no strong evidence for any consistent behavioral effects except for a few data points that suggested improvement and some that suggested further impaired behavior. Fingolimod is predominantly used to treat MS by reducing inflammation and slowing disease progression, which may help preserve motor function. Still, the evidence does not conclusively show that it directly improves locomotor deficits. However, reducing the expression of mutant *Atxn1* in SCA1 mice resulted in behavioral recovery, which was associated with a significant decrease in microglial density and TNF-α levels [51,52]. Hence, a more optimal inhibition of the immune response may also positively affect the behavioral issues.

Is there a way to optimize the therapeutic potential of these interventions on the S1P pathway? Although fingolimod inhibits both Sphk1 and Sphk2 activity, it is predominantly phosphorylated in vivo by the Sphk2 enzyme [53,54], suggesting that fingolimod may not be the most selective option to target Sphk1. The activity of Sphk1 and Sphk2 seems redundant, but this has not been fully explored, so potential subtype-specific effects may exist. We could not test for a potential behavioral improvement in the SCA1 mice in which we deleted the *Sphk1* gene due to the complex breeding scheme that yielded a very low number of the required mutant mice. Thus, the therapeutic potential could perhaps be improved by a more selective blocking of Sphk1. Recently, more selective Sphk1 inhibitors have been developed, including SK-I [55,56] and PF-543 [57,58], but these have not been tested in the clinical setting. Alternatively, other S1P-receptor modulators have been developed, including ozanimod and ponesimod, that more specifically target S1PR1 and 5, or only S1PR1, respectively. Given that the recovery effect in the current experiment was incomplete, combining therapy with other recently developed treatment options could be the most promising solution. These options include genetic approaches, e.g., ASOs and modulation of neuronal excitability, currently tested in phase III clinical trials using troriluzole [59,60]. Interestingly, some pathological features that fingolimod could not address, e.g., molecular layer thickness and balance beam performance, are affected when the activity of cerebellar PCs is explicitly manipulated [61]. This suggests that there may be two components to disease pathogenesis: an excitability component and a neuroinflammatory component. If this hypothesis holds, a therapy combining troriluzole and an anti-inflammatory drug like fingolimod would potentially reduce disease progression even more.

A second important factor is the timing of treatment. In the second experiment (Figure 4), we found a surprising increase in the behavioral deficit of SCA1 mice during adolescence and early adulthood. This suggests that the moderate immune response of astrocytes and microglia in the early stage of the disease may help improve neuronal functioning [62,63]. Early activation of astrocytes, measured by changes in GFAP expression, may initially play a protective role by attempting to maintain homeostasis against neuronal stress. We did not observe a negative effect when fingolimod was administered from even younger, at 2 weeks of age, suggesting a potentially even more complex optimal treatment course. Taken together, a comprehensive understanding of the role of the S1P pathway in neurodegenerative disorders is critical for future therapeutic approaches.

There are a few relevant limitations to our study. First, the mouse is the most common model to study (dys)functioning of the brain, and although the brain is organized mainly in the same manner, there are significant differences that could mean results cannot be translated to human patients. Apart from differences in genetic elements, an essential component in neurodegenerative models is that aging and disease progression are very different in humans. Second, our studies used the ATXN1[82Q] mouse model, expressing the mutated *Atxn1* only in PCs. Hence, fingolimod treatment could have different effects if all cells express the mutated gene. It is difficult to say whether this would positively or negatively impact the therapeutic effect of fingolimod.

In summary, neuroinflammation commonly plays a critical role in the pathogenesis of neurodegenerative disorders, including most spinocerebellar ataxias. Spinocerebellar ataxia type 1 (SCA1) is no exception, as is demonstrated by the activation of glial cells, particularly astrocytes and microglia. Our results indicate that while early immune responses mediated by astrocytes and microglia may initially exert neuroprotective effects, excessive inflammatory activity can be inhibited using fingolimod. Still, this treatment option has limited impact on the progression of the behavioral phenotype.

## 4. Methods and Materials

### 4.1. Animals

All animals in this study were handled and kept under conditions that respected the guidelines of the Dutch Ethical Committee for Animal Experiments and experiments were performed according to the Institutional Animal Care and Use Committee of Erasmus Medical Center (IACUC Erasmus MC), European and Dutch National Legislation. The Institutional Welfare Committee of Erasmus Medical Center approved all experimental protocols. In this study, we used Tg(Pcp2-ATXN1*82Q)5Horr mice, referred to as ATXN1[82Q]. Experiments were performed on heterozygous transgenic mice of both sexes, expressing *Atxn1* with an 82 CAG repeat expansion under the control of the *Pcp2* promoter, thereby restricting mutant ATXN1 expression specifically to cerebellar Purkinje neurons. Mice were bred in an FVB/NHsd genetic background, and wild-type littermates were used as controls. The length of the terminal CAG repeat across generations was verified by PCR using the primers Rep1 (5′-AAC TGG AAA TGT GGA CGT-3′) and Rep2 (5′-CAA CAT GGG CAG TCT GAG-3′), described previously [17]. All mutant mice exhibited cerebellar neuropathy and motor and behavioral abnormalities consistent with the SCA1 phenotype. Mice were housed under standard laboratory conditions at 22–24 °C, 30–60% relative humidity, and a 12:12 h light/dark cycle (light on from 7 AM to 7 PM). Experiments were performed during the light period. Commercially available pellet diets and water were available ad libitum. Mice were randomly housed with two to four siblings of the same sex, mixing mutants and controls.

### 4.2. Fingolimod Administration

A saline vehicle containing 0.2% DMSO was used for intraperitoneal (i.p.) injection. Each mouse’s body weight was measured, and injections were administered at a quantity of 10 μL per gram. Fingolimod (FTY720, Sigma-Aldrich, Amsterdam, The Netherlands) was prepared in the vehicle at 10 mg/mL, resulting in a final 1 mg/g dosage. We injected the animals i.p. three times per week with a one-day interval (Mo-We-Fr).

To evaluate the therapeutic effects of fingolimod across different stages of disease progression, three distinct treatment regimens were implemented:Regimen 1. Treatment for 8 weeks, starting week 2, behavioral analysis from week 4;Regimen 2. Treatment for 8 weeks, starting week 5, behavioral analysis from week 5;Regimen 3. Treatment for 4 weeks, starting week 10, behavioral analysis from week 10.

All treatments were administered according to the same dosing protocol, and vehicle-injected littermate wild-type mice served as controls for each experimental group.

### 4.3. Balance Beam and Rotarod Tests

Balance and motor coordination were assessed using the balance beam test. The test consists of two elevated platforms connected by 6 mm or 12 mm diameter circular metal beams. Mice were placed on the first platform and had to cross the beam to reach the second platform and, thereby, their home cage. On each experiment day, mice crossed the 12 mm beam first and subsequently the 6 mm beam, each three times. All mice received one day of training on both beams, before the first experiment days. Our dependent variables are the time it takes for mice to cross the beams and the number of foot slips in a 50 cm stretch of the beam marked by red tape.

Motor coordination and motor learning were also assessed using the accelerating rotarod test. The rotarod (Ugo Basile, Gemonio, Italy) consists of an accelerating rod with five compartments and a monitoring device. Mice must walk on the rod while it accelerates from 4 rpm to 40 rpm over a 300 s period; the maximum walking time is set to 300 s. The monitoring device automatically records the time the mouse falls off the rotating rod or completes the test. When the animal stopped walking, held onto the rod, and rotated thrice in a row, it was also considered a “fall”. The mice were tested on rotarod tests once a week, and our dependent variable was the latency of falling from the rod. All behavior experiments were done in one week with a one-day interval.

### 4.4. Tissue Collection

Animals were deeply anesthetized by intraperitoneal injection of sodium pentobarbital (60 mg/kg), rinsed with saline, and then perfused with 4% paraformaldehyde (PFA) in 0.12 M phosphate buffer (PB) (pH 7.6). Brains were fixed in 4% PFA for 1 h at room temperature (rT) and then transferred to a 10% sucrose solution at 4 °C overnight. The pia was removed under a microscope, and the brain was placed in a jar with 10% sucrose. The brain was embedded in a 14% gelatin/10% sucrose mixture, and gelatin blocks were fixed in 30% sucrose/10% formaldehyde at rT for 2 h and then stored in 30% sucrose at 4 °C overnight. Subsequently, coronal sections were cut to 50 μm thickness using a freezing microtome, and the slices collected in 0.1 M PB.

### 4.5. Immunohistochemistry

Free-floating sections were rinsed with 0.1 M PB and then incubated in 10 mM sodium citrate (pH6) at 80 °C for 2 h for antigen retrieval. For immunofluorescence free-floating sections, sections were rinsed with 0.1 M PB and then washed three times in phosphate-buffered saline (PBS) for 10 min each. The sections were then blocked in PBS/0.5% TritonX-100/10% normal horse serum (NHS) solution at rT for 60 min, using the following primary antibody dilutions: Calbindin 1:20,000 (rabbit polyclonal antibody, CB-38a; Swant, Burgdorf, Switzerland), GFAP 1:1000 (chicken polyclonal antibody, 4674; Abcam, Cambridge, UK), Iba1 1:2000 (rabbit polyclonal antibody, 019-19741; WAKO, Neuss, Deutschland), and VGLUT2 (1:2000; guinea pig polyclonal antibody, AB2251-I, Millipore, Germany). After rinsing three times with PBS for 10 min each, the sections were incubated for 2 h at rT in a PBS/0.4% Triton-X 100/2% NHS solution containing secondary antibodies conjugated to Alexa 488 (1:400, Jackson ImmunoResearch, West Grove, PA, USA), Cy3 (1:400, Jackson ImmunoResearch, West Grove, PA, USA), washing twice with 0.1 M PB for 5 min. After a 10-min incubation with DAPI (diluted 1:1000; Jackson ImmunoResearch, West Grove, PA, USA), the samples were rinsed twice with 0.1 M PB with each wash lasting 5 min. The sections were mounted on coverslips in gelatin/chrome alum solution and covered with Mowiol (Polysciences Inc., Darmstadt, Germany).

### 4.6. Imaging and Quantification

Fluorescence example images (1024 × 1024 pixels; 20× magnification) were acquired with the LSM 700 confocal microscope (Carl Zeiss Microscopy), and fluorescent images (10× magnification) for quantification were generated using the Axio Imager.M2 (both Zeiss, Oberkochen, Germany). GFP-positive, Iba1-positive, calbindin-positive, and VGLUT2 cells were quantified using a cell counter and intensity analysis with Zen blue 3.10 software (Carl Zeiss, Oberkochen, Germany). Iba1-positive cell counts were acquired from the anterior cerebellum; all other analyses were performed using a selection tool to select regions of interest across all coronal cerebellar sections.

### 4.7. Statistical Analysis

Statistical analysis was performed using GraphPad Prism version 9.4.1. Two-way analysis of variance (ANOVA) was used to assess differences between treatment groups when data were normally distributed, as determined by the Shapiro–Wilk test. When data were not normally distributed, a mixed model was used. A *p*-value < 0.05 was considered statistically significant. All experimental procedures and analyses were conducted with the experimenters blinded to the genotype and treatment groups. Microglia counting was independently performed by two investigators to confirm reproducibility. Some panels were created with BioRender (https://www.biorender.com), licensed to M. Schonewille.

## Figures and Tables

**Figure 1 ijms-26-04698-f001:**
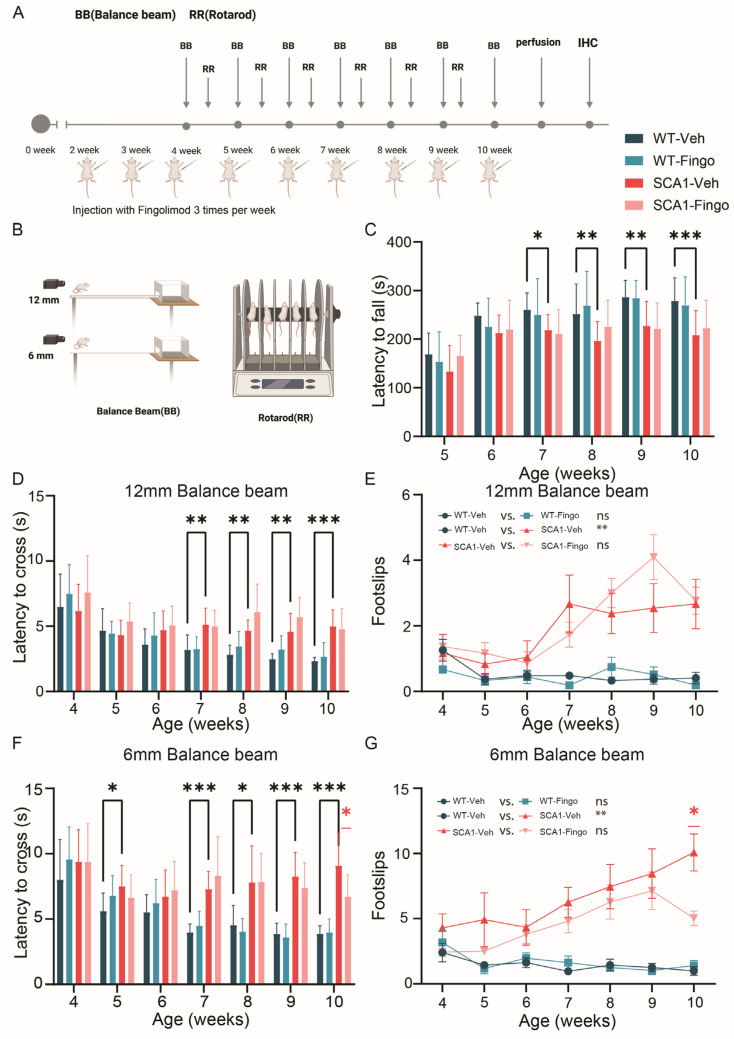
Fingolimod treatment has minimal beneficial effects on impairment in the locomotion of SCA1 mice. ATXN1[82Q] mutant and littermate control mice were injected with fingolimod or vehicle solution three times a week, starting from 2 weeks of age, and behavioral experiments were initiated from 5 weeks of age (**A**). Rotarod and balance beam (**B**) were used to track cerebellar ataxia’s development and potential recovery. Analyzed parameters include rotaod latency to fall (**C**), 12 mm balance beam latency to cross (**D**), number of foot slips (**E**), 6 mm balance beam latency to cross (**F**), and number of foot slips (**G**). Experimental groups consisted of nine control mice injected with vehicle (WT-Veh), nine control mice injected with fingolimod (WT-Fingo), 10 ATXN1[82Q] mice injected with vehicle (SCA1-Veh), and eight ATXN1[82Q] mice injected with fingolimod (SCA1-Fingo). The non-normally distributed rotarod, the fall latency on 12 mm BB, and foot slips on the 12 mm BB were analyzed using a mixed-effects model to compare WT-Veh with SCA1-Veh. SCA1-Veh and SCA1-Fingo were normally distributed. Therefore, a two-way ANOVA was used to assess the effect of genotype and treatment. For data that conform to a normal distribution, the fall latency on 6 mm BB, foot slips on the 6 mm BB, and a two-way ANOVA were used. All values are presented as mean ± sem.* *p* < 0.05, ** *p* < 0.01 and *** *p* < 0.001; only the outcomes for WT-Veh vs. SCA1-Veh, WT-Veh vs. WT-Fingo and SCA1-Veh vs. SCA1-Fingo are indicated, ns—not significant.

**Figure 2 ijms-26-04698-f002:**
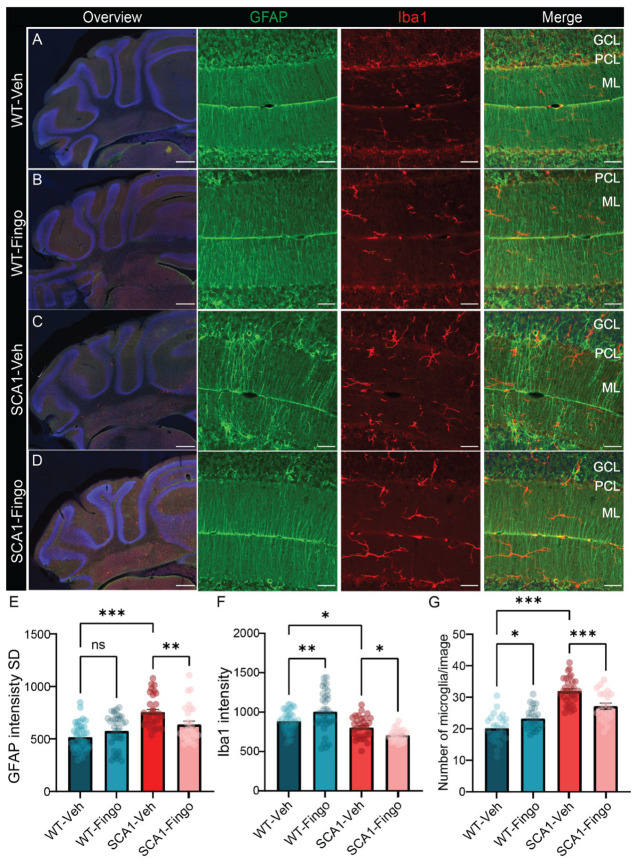
Fingolimod attenuates the inflammatory responses in ATXN1[82Q] mice. Overview micrographs of GFAP and Iba1 immunofluorescent labeling of the posterior cerebellum of SCA1 mice (**C**,**D**) and control littermate mice (**A**,**B**) at week 10. Higher-resolution confocal images show GFAP and Iba1 staining in the anterior cerebellum (lobule IV). (**E**) Analysis of the standard deviation of the intensity of GFAP labeling measured across the anterior and posterior cerebellar cortex. (**F**) Analysis of Iba1 intensity in the same sections. (**G**) Analysis of numbers of microglia in the anterior cerebellum per analyzed square. For E and F, each experimental group consisted of five mice, and for each mouse, we analyzed the entire cerebellar cortex in five sections. For (**G**), we analyzed at least three regions of interest per mouse for five mice per group. ML = molecular layer; PCL = Purkinje cell layer; GCL = granule cell layer. Scale bars: overview image, 500 μm; insets, 40 μm. For data that conform to a normal distribution, a two-way ANOVA was used. All values are presented as mean ± sem, circles indicate individual sample points. * *p* < 0.05, ** *p* < 0.01, and *** *p* < 0.001; only the outcomes for WT-Veh vs. SCA1-Veh, WT-Veh vs. WT-Fingo and SCA1-Veh vs. SCA1-Fingo are indicated, ns—not significant.

**Figure 3 ijms-26-04698-f003:**
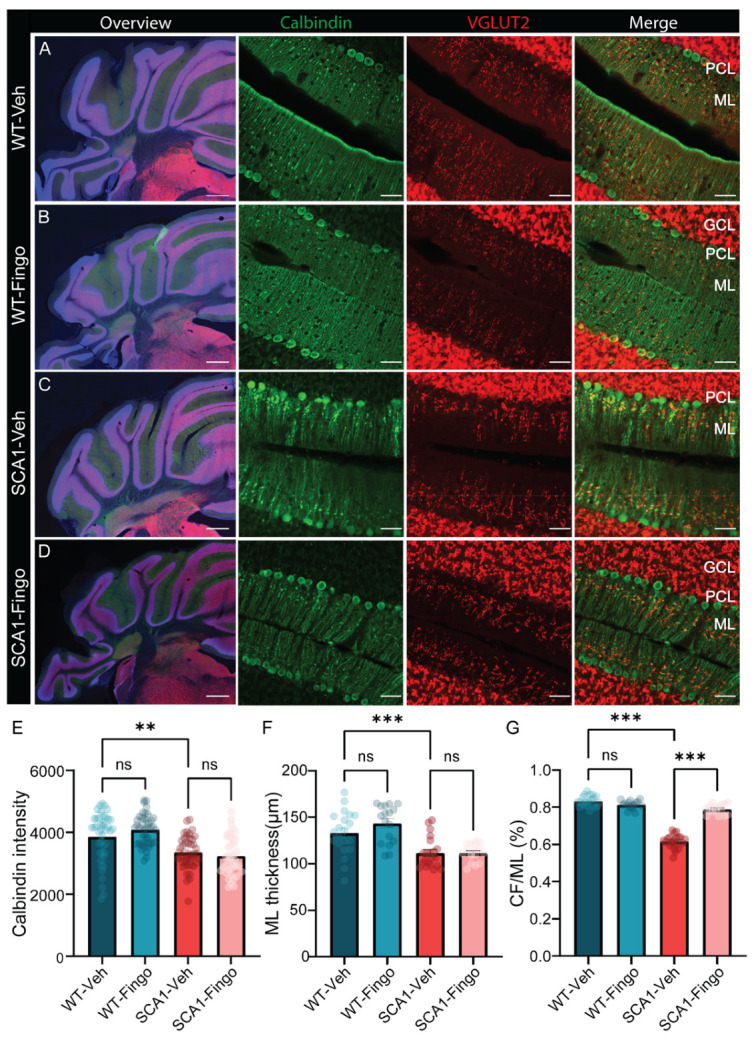
Fingolimod ameliorates the reduced input area of climbing fibers in SCA1 mice. Representative micrographs of the anterior cerebellar cortex with immunofluorescent labeling of VGLUT2 and Calbindin (green) of SCA1 mice (**C**,**D**) and WT mice (**A**,**B**), sacrificed at 10 weeks of age. Detailed images of calbindin and VGLUT2 are in lobule IV. Analysis of calbindin density (**E**), molecular layer thickness (**F**), and climbing fiber coverage of ML (CF/ML%), panel (**G**). ML = molecular layer; PCL = Purkinje cell layer; GCL = granule cell layer. Scale bars are 500 μm for the overview image on the left and 40 μm for the magnifications. Each experimental group consisted of five mice, and for each mouse, for calbindin intensity analysis, five sections were analyzed. For ML thickness and CF/ML, five sections were analyzed for each mouse, and we included the average value per section. All values are presented as mean ± sem ** *p* < 0.01 and *** *p* < 0.001. All data were tested for significance using a two-way ANOVA or mixed-model test; only the outcomes for WT-Veh vs. SCA1-Veh, WT-Veh vs. WT-Fingo and SCA1-Veh vs. SCA1-Fingo are indicated, ns—not significant.

**Figure 4 ijms-26-04698-f004:**
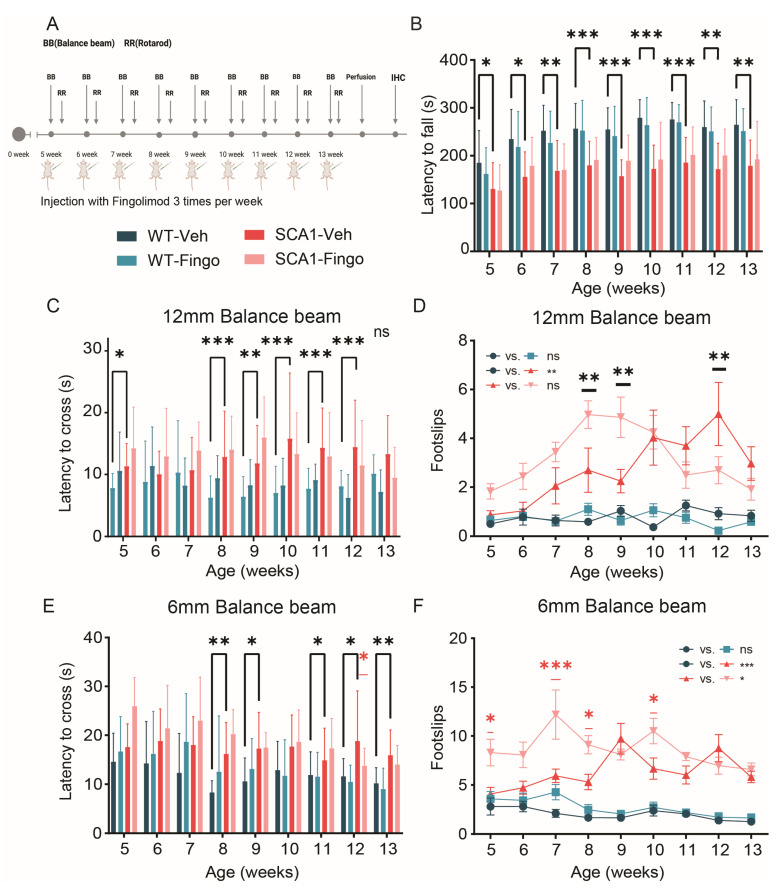
Delaying the start of treatment with fingolimod has bidirectional effects on behavioral impairment in SCA1 mice. (**A**) ATXN1[82Q] mutant and littermate control mice were injected with fingolimod or vehicle solution three times a week, starting from 5 weeks of age, and behavioral experiments began at 5 weeks. We analyzed rotarod performance (**B**), 12 mm balance beam latency to cross (**C**), the number of foot slips (**D**), the 6 mm balance beam latency to cross (**E**), and the number of foot slips (**F**). All experimental groups consisted of 12 mice. The non-normally distributed rotarod results were analyzed using a mixed-effects model to compare WT-Veh with SCA1-Veh and SCA1-Veh with SCA1-Fingo. For normally distributed data—crossing latency on 12 mm BB, foot slips on 12 mm BB, crossing latency on 6 mm BB, and foot slips on 6 mm BB—a two-way ANOVA was used to assess the effect of genotype and treatment between WT-Veh vs. SCA1-Veh, WT-Veh vs. WT-Fingo and SCA1-Veh vs. SCA1-Fingo. All values are presented as mean ± sem.* *p* < 0.05, ** *p* < 0.01 and *** *p* < 0.001; only the outcomes for WT-Veh vs. SCA1-Veh, WT-Veh vs. WT-Fingo and SCA1-Veh vs. SCA1-Fingo are indicated, ns—not significant.

**Figure 5 ijms-26-04698-f005:**
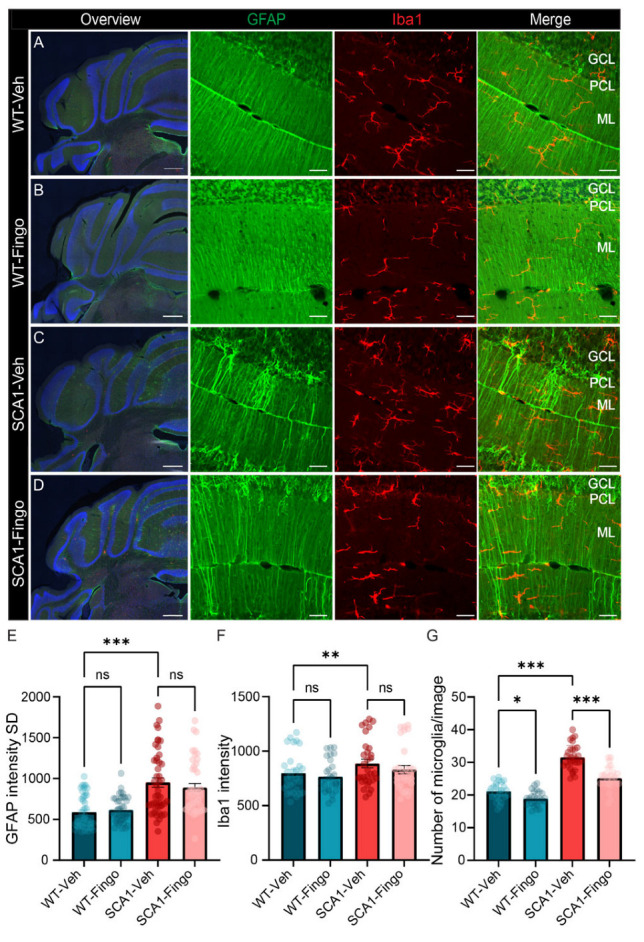
Long-term use of fingolimod reduces GFAP expression and the number of microglia in SCA1 mice. Overview micrographs of GFAP and Iba1 immunofluorescent labeling of the posterior cerebellum of SCA1 mice (**C**,**D**) and control littermate mice (**A**,**B**) at week 13. Higher-resolution confocal in the overview images shows GFAP and Iba1 staining in lobule IV. (**E**) Analysis of the standard deviation of the intensity of GFAP labeling measured across the cerebellar cortex. (**F**) Analysis of Iba1 intensity in the same sections. (**G**) Analysis of the number of microglia in the anterior cerebellum. Each experimental group consisted of five mice, and five sections were analyzed for each mouse. ML = molecular layer; PCL = Purkinje cell layer; GCL = granule cell layer. Scale bars are 500 μm for the overview image on the left and 40 μm for the magnifications. All values are presented as mean ± sem and a two-way ANOVA was used to test for significance. * *p* < 0.05, ** *p* < 0.01 and *** *p* < 0.001; only the outcomes for WT-Veh vs. SCA1-Veh, WT-Veh vs. WT-Fingo and SCA1-Veh vs. SCA1-Fingo are indicated, ns—not significant.

**Figure 6 ijms-26-04698-f006:**
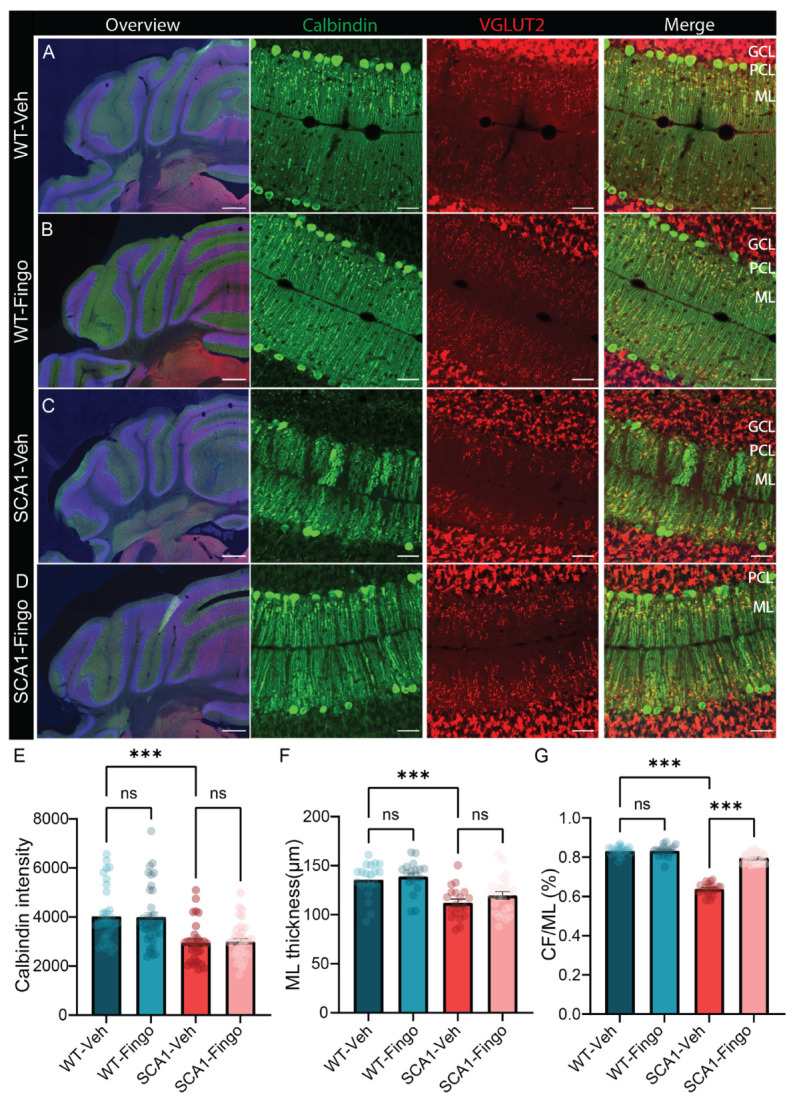
Fingolimod increases the input of climbing fibers in SCA1 mice in the later disease stage. Representative micrographs of PCs and expression in the posterior cerebellum of SCA1 mice (**C**,**D**) and WT mice (**A**,**B**) at week 13 (scale bar = 200 μm). Detailed images of calbindin and VGLUT2 in SCA1 mice and WT mice in lobule VII at week 13 (scale bar = 50 μm). ML = molecular layer; PCL = Purkinje cell layer; GCL = granule cell layer. Four mice were in each group. Analysis of calbindin density (**E**), molecular layer thickness (**F**), and CF/ML% (**G**). ML = molecular layer; PCL = Purkinje cell layer; GCL = granule cell layer. Scale bars are 500 μm for the overview image on the left and 40 μm for the magnifications. Each experimental group consisted of five mice. For calbindin intensity analysis, five sections were analyzed. For ML thickness and CF/ML, five sections were analyzed in each mouse, and we calculated the average value. All values are presented as mean ± sem and a two-way ANOVA was used to test for significance. *** *p* < 0.001; only the outcomes for WT-Veh vs. SCA1-Veh, WT-Veh vs. WT-Fingo and SCA1-Veh vs. SCA1-Fingo are indicated, ns—not significant.

**Figure 7 ijms-26-04698-f007:**
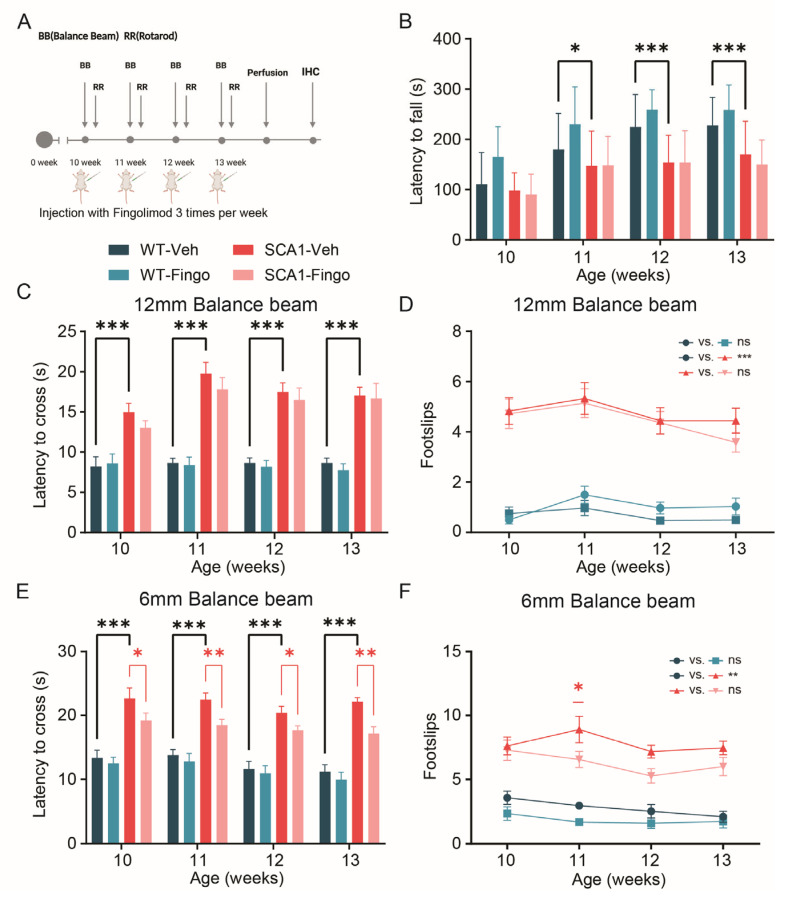
Short-term fingolimod administration minimally improves behavioral deficits in SCA1 mice in later stages of disease development. (**A**) ATXN1[82Q] mutant and littermate control mice were injected with fingolimod or vehicle solution three times a week for four weeks, starting from 10 weeks of age, combined with behavioral experiments. Rotarod (**B**), 12 mm balance beam latency (**C**) and number of foot slips (**D**), and 6 mm balance beam latency to cross (**E**) and number of foot slips (**F**). All experimental groups consisted of 12 mice per group. Rotarod and latency of 12 mm BB were not normally distributed and therefore were analyzed using a mixed-effects model to compare WT-Veh with SCA1-Veh and SCA1-Veh with SCA1-Fingo. For normal distribution data, foot slips on the 12 mm BB, crossing latency on the 6 mm BB, and foot slips on the 6 mm BB, a two-way ANOVA was used to assess the effect of genotype and treatment. All values are presented as mean ± sem.* *p* < 0.05, ** *p* < 0.01 and *** *p* < 0.001; only the outcomes for WT-Veh vs. SCA1-Veh, WT-Veh vs. WT-Fingo and SCA1-Veh vs. SCA1-Fingo are indicated, ns—not significant.

**Figure 8 ijms-26-04698-f008:**
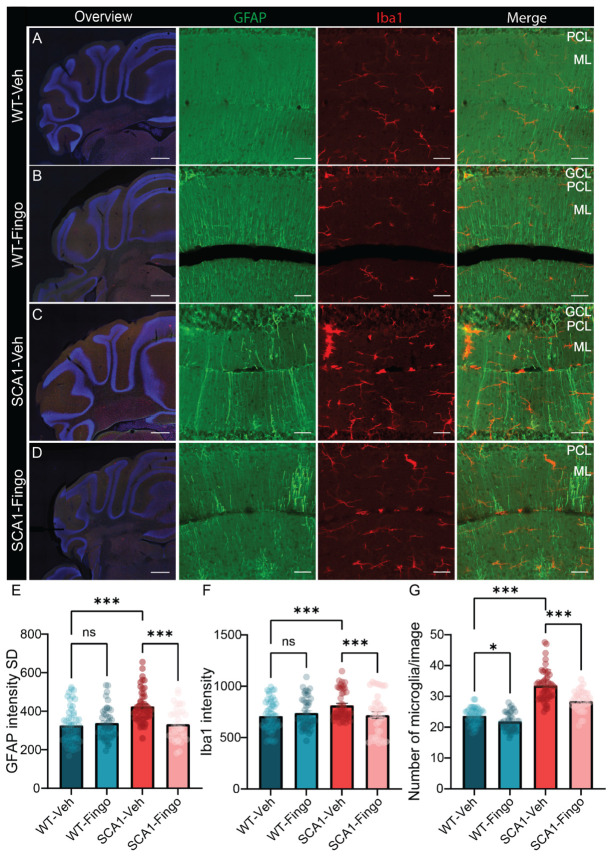
Short-term use of fingolimod is sufficient to reduce the immune responses in SCA1 mice. Overview micrographs of GFAP and Iba1 expression in the posterior cerebellum of SCA1 mice (**C**,**D**) and WT mice (**A**,**B**) at week 13 (scale bar = 500 μm). Detailed confocal images of GFAP and Iba1 in lobule IV in SCA1 mice and WT mice at week 13 (scale bar = 40 μm). Analysis of GFAP intensity variance (**E**), Iba1 intensity (**F**), and microglia count in the anterior cerebellum (**G**). ML = molecular layer; PCL = Purkinje cell layer; GCL = granule cell layer. Each group consists of five mice, and five sections were chosen to be analyzed. All values are presented as mean ± sem.* *p* < 0.05, and *** *p* < 0.001; two-way ANOVA was used and only the outcomes for WT-Veh vs. SCA1-Veh, WT-Veh vs. WT-Fingo and SCA1-Veh vs. SCA1-Fingo are indicated, ns—not significant.

**Figure 9 ijms-26-04698-f009:**
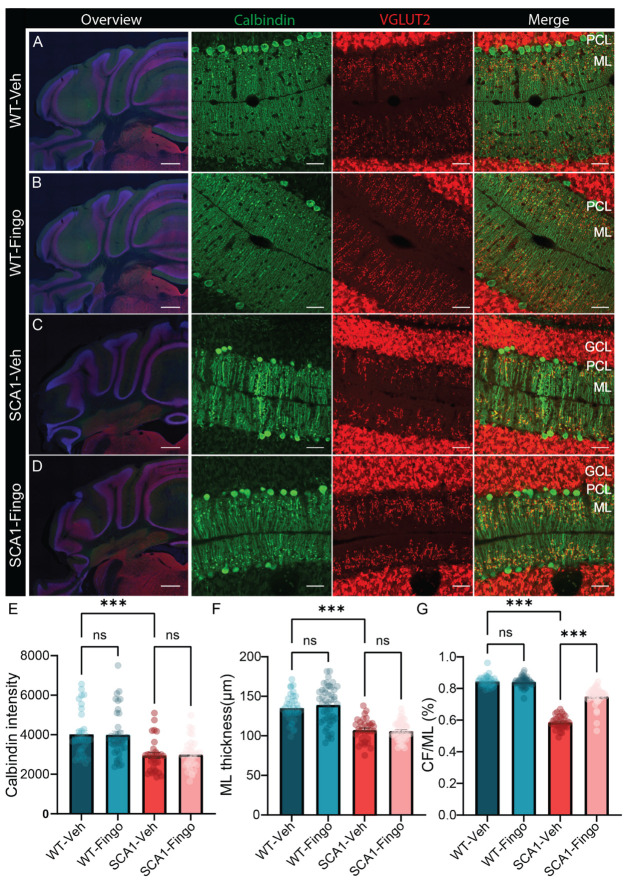
Short-term fingolimod treatment ameliorates the reduced input of climbing fibers in SCA1 mice. Representative micrographs of PC and expression in the posterior cerebellum of SCA1 mice (**C**,**D**) and WT mice (**A**,**B**) at week 13 (scale bar = 500 μm). Detailed images of calbindin, VGLUT2 in lobule IV of SCA1 mice and WT mice at week 13 (scale bar = 40 μm). Analysis of calbindin density (**E**), molecular layer thickness (**F**), and CF/ML% (**G**). ML = molecular layer; PCL = Purkinje cell layer; GCL = granule cell layer. Each group consists of five mice. For calbindin intensity analysis, five sections were analyzed; for ML thickness and CF/ML, five sections were analyzed from each mouse, and we calculated the average value. All values are presented as mean ± sem and a two-way ANOVA was used to test for significance. *** *p* < 0.001; only the outcomes for WT-Veh vs. SCA1-Veh, WT-Veh vs. WT-Fingo and SCA1-Veh vs. SCA1-Fingo are indicated, ns—not significant.

## Data Availability

All data are available upon request.

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
