# Peer review of "Fingolimod Prevents Neuroinflammation but Has a Limited Effect on the Development of Ataxia in a Mouse Model for SCA1"

_ijms, 2025, doi:10.3390/ijms26104698_

Round 1
Reviewer 1 Report
Comments and Suggestions for Authors
Overall comments:
This paper describes a study to test the efficacy of an inhibitor of neuroinflammation on spinocerebellar ataxia disease progression in a moue model. The treatments had variable effects on measures of neuroinflammation depending on treatment regimen, but did not have consist beneficial effects with respect to behavioral phenotypes. Overall, the paper does provide strong support for role of neuroinflammation in the disease process.
Specific Comments:
Introduction
Lines 37-39: provide citations to papers describing the disease pathophysiology.
Lines 58-60: The investigators indicate that they chose to use a mouse model that only expresses the risk variant of ATXN1 in Purkinje cells. They need to indicate the limitations of this model in predicting results in human subjects in which all cell types would have the variant genotype.
Lines 72-23: The study is based on the hypothesis that neuroinflammation plays a substantial role in the disease pathology. They need to cite evidence that supports this hypothesis.
A rational needs to provided for designing the experiments to test treatments started at different ages and for different durations.
Materials and Methods
Although this mouse model was developed and characterized previously, it would be helpful to cite the evidence that the transgene expression is restricted to the Purkinje cells.
Lines 126-131: In the abstract it is stated that mice were treated with fingolimod “during three different periods.” It is not clear what these periods were. Was treatment started at the same age for all mice and the duration of treatment varied, or was the treatment started at different ages and the duration of treatment kept the same? This information can be gleaned from the results but these details need to be provided in the Methods section.
The quantitative analyses should have been performed masked to avoid bias.
Results
Paragraph 3.1. As indicated above, it is not clear whether treatment was started at the same age for all mice and the duration of treatment varied, or whether the treatment started at different ages and the duration of treatment was kept the same. It is difficult to interpret the results without this information.
Figure 2. In the vehicle-treated SCA1 mice, the Iba1 labeling was not elevated relative to the control mice, suggesting that the microglia are not involved in the neuroinflammatory response in this disease. However, the number of microglia was increased. These data appear to be contradicatory.
Discussion
Lines 488-491: “More specifically, long-term treatment with fingolimod starting from the early stages of the disease was associated with an increase in behavioral deficits, whereas shorter treatment with fingolimod starting at a later stage significantly alleviated part of the locomotion impairment.” There needs to be some discussion of what the mechanisms for this might be. This related back to the question of what the rationale was for using 3 different treatment regimens. It is not clear what hypothesis was being tested by varying the treatment regimen.
The authors misuse the term “ultrastructure”. Ultrastructure refers to electron microscopy which was not done in this study.
Author Response
Introduction
Lines 37-39: provide citations to papers describing the disease pathophysiology.
We have added citations (line 38)
Lines 58-60: The investigators indicate that they chose to use a mouse model that only expresses the risk variant of ATXN1 in Purkinje cells. They need to indicate the limitations of this model in predicting results in human subjects in which all cell types would have the variant genotype.
We have added explanations in the introduction and methods to more clearly indicate the nature of the mouse model used here (lines 60-64, 127-128).
Lines 72-73: The study is based on the hypothesis that neuroinflammation plays a substantial role in the disease pathology. They need to cite evidence that supports this hypothesis.
We have added citations to support the involvement of inflammatory mechanisms in neurodegenerative diseases (line 78).
A rational needs to provided for designing the experiments to test treatments started at different ages and for different durations.
Based on this comment and that of reviewer 3, we now explain the treatment regimens and the rationale behind them more extensively, eg in the introduction (lines 109-112), the methods (lines 144-150) and Discussion (lines 518-522). These are in addition to the text explaining the rationale for the selected treatment regimens at the start of paragraph 3.1 (regimen 1: lines 220-227), paragraph 3.3 (regimen 2: lines 346-349) and paragraph 3.5 (regimen 3: lines 439-446).
Materials and Methods
Although this mouse model was developed and characterized previously, it would be helpful to cite the evidence that the transgene expression is restricted to the Purkinje cells.
We now explain that the transgene is exclusively expressed by Purkinje cells and include the citation in the methods (lines 127-128).
Lines 126-131: In the abstract it is stated that mice were treated with fingolimod “during three different periods.” It is not clear what these periods were. Was treatment started at the same age for all mice and the duration of treatment varied, or was the treatment started at different ages and the duration of treatment kept the same? This information can be gleaned from the results but these details need to be provided in the Methods section.
We have added the following text to the Methods section to explain the treatment paradigms used:
“Treatment regimens:
We tested the effects of Fingolimod using three different treatment regimes:
1. Mice treated for 8 weeks, starting from week 2
2. Mice treated for 8 weeks, starting from week 5
3. Mice treated for 4 weeks, starting from week 10”
The quantitative analyses should have been performed masked to avoid bias.
This was indeed the case; we included this in a statement in the Methods section (lines 210-213).
Results
Paragraph 3.1. As indicated above, it is not clear whether treatment was started at the same age for all mice and the duration of treatment varied, or whether the treatment started at different ages and the duration of treatment was kept the same. It is difficult to interpret the results without this information.
As indicated in response to this reviewer comments on the introduction above, we have now included additional statements about the experimental design in the introduction (lines 112-116), the methods (lines 147-153) and Discussion (lines 525-529), in addition to the schematic drawing in the first panels of Figures 1, 4 and 7.
Figure 2. In the vehicle-treated SCA1 mice, the Iba1 labeling was not elevated relative to the control mice, suggesting that the microglia are not involved in the neuroinflammatory response in this disease. However, the number of microglia was increased. These data appear to be contradicatory.
This is indeed an interesting point, for which we can only speculate that it has to the ability of certain parameters to represent progression of pathological changes. We addressed this now with the following sentence (lines 288-291):
“The apparent contradiction between the intensity and number of microglia could be caused by the fact that the changes in intensity occur later and are a more indirect measurement, whereas the number of microglia more directly reflects changes in glia.“
Discussion
Lines 488-491: “More specifically, long-term treatment with fingolimod starting from the early stages of the disease was associated with an increase in behavioral deficits, whereas shorter treatment with fingolimod starting at a later stage significantly alleviated part of the locomotion impairment.” There needs to be some discussion of what the mechanisms for this might be. This related back to the question of what the rationale was for using 3 different treatment regimens. It is not clear what hypothesis was being tested by varying the treatment regimen.
The rationale for the selected treatments regimens is presented in the first sentences of paragraphs 3.1, 3.3. and 3.5. In addition, we now briefly indicate the different treatment regimens in the first paragraph of the discussion (lines 5118-522).
We agree with the reviewer that the mechanism underlying the difference in starting treatment early vs. later is very interesting. We now address this in the discussion as follows (lines 540-548):
“One potential explanation for this difference could be that early in the disease, inflammation may serve an adaptive or compensatory role, and its suppression could interfere with protective astrocytes responses[44-45]. In contrast, at later stages, neuroinflammation then potentially shifts towards a maladaptive, chronic response characterized by gliosis and neurotoxicity, in which fingolimod’s modulation of S1P signaling may help restore homeostasis and limit damage. Interestingly, this stage-dependent shift in inflammatory function has been propose before, in SCA1 treatment [44], and thus putatively underlies the differential effects observed across treatment windows.”
The authors misuse the term “ultrastructure”. Ultrastructure refers to electron microscopy which was not done in this study.
We apologize for the mistake and have replaced “ultrastructure” with “morphological”

Reviewer 2 Report
Comments and Suggestions for Authors
The authors evaluated the treatment of Fingolimod in the Spinocerebellar ataxia type 1 (SCA1) animal model based on previous findings showing that inhibiting the S1P pathway and the associated reduction of inflammatory reactions could be a potential therapeutic approach. There is currently no accepted treatment option for SCA1-dependent pathology, although antisense oligonucleotides (ASOs) that inhibit the expression of the mutant Atxn1 gene have been developed, but phase III trials still must be completed. Here they focused on the cerebellum-behaviour changes of mice like motor coordination and balance, they used the ATXN1[82Q] SCA1 mouse model, which expressed the mutated Atxn1 only in PC testing the hypothesis that fingolimod treatment can alleviate motor dys function and inflammatory responses in an SCA1 mouse model by inhibiting the S1P pathway.
The topic is interesting and the results are sound. Images are clear and Statistical analyses is well performed. I accept the manuscript as it is
Author Response
We sincerely thank the reviewer for their kind and encouraging comments.

Reviewer 3 Report
Comments and Suggestions for Authors
see attached file

Author Response
Line 76 After sphingolipid-1P, please remove the abbreviation S1P.
We have fixed the double abbreviation (line 80).
Line 85 It would be useful for the reader to clarify the role of SPHK1 here (although this role will be described later).
We agree and have linked S1P to Sphk1 here now (line 83).
Line 195 The sentence is poorly worded and suggests that fingolimod is an SPHK1 inhibitor, whereas it seems to me that the compound is known to inhibit S1P receptors.
Although fingolimod is sometimes referred to as an SPHK1 inhibitor, we kept it more general and refer to it as S1P-modulator here now.
In the graphs of footprints versus age (Figures 1, 4 and 7) it is not clear which values are significantly different. Perhaps a different symbol should be used for each comparison and one, two or three symbols should be used depending on the p-value. Also, why are there two stars on Fig 1E ? and on Fig 1F ? See also Fig 7.
We understand the confusion and now use black asterisks for WT-Veh vs. SCA1-Veh and red asterisks for SCA1-Veh vs. SCA1-Fingo, with different levels of significance related to the number of asterisks. We are not sure we understand the point of the reviewer regarding Fig. 1E / F / 7.
Line 255-258 It is not clear to me why the authors measure the variance and intensity of fluorescence obtained with Iba1 (which labels microglia) on the posterior cerebellum and count the number of microglial cells on the anterior cerebellum.
We understand the confusion. In previous work, we found a prominent increased in the anterior part of the cerebellum that could be rescued by deletion of SPHK1, so we counted microglia in the anterior cerebellum. For the quantification of the fluorescence intensity, we aligned the analysis with that of GFAP and analyzed both posterior and anterior cerebellar sections. We now include Supplementary Figures 2,3,4 in which we show the counts for anterior and posterior separately.
Line 285 Fig 2 caption : the boxed areas do not appear on the pictures ! Same thing Line 376. Also : «Analysis of numbers of microglia in the anterior cerebellum per analyzed square » where are these squares ?
That was indeed an incorrect statement in the legends, we apologies for the mistake. The overview images are from a slide scanner (Zeiss Axio Imager.M2) and the magnifications are confocal images (LSM700). We adjusted the text in the legends from "Magnifications of the boxed area…" to "Higher resolution confocal image…".
Line 318-322 The authors describe here the 12mm balance beam test in which fingolimod treatment has no effect (Figure 4C) "except for a single significant effect observed at week 12 (line 321)", but this difference appears in the 6mm balance beam test (Figure 4E).
We have fixed this error, and thank the reviewer for pointing it out (lines 358-359 and 360-361).
Line 389 389 Fig. 6 caption, if I understand correctly, it was in week 13 and not week 10 that the pictures were obtained.
Indeed, we have corrected it.
Line 408 For the rotarod test shown in Figure 7B, is it not surprising that no difference was observed between SCA1 and WT mice at 10 weeks?
This is an interesting point. In order to verify the therapeutic effect of fingolimod over up to an eight-week period, we used a longitudinal approach to track the changes in cerebellar function. To avoid over-training mice, we limited the number of sessions to three per week, significantly less than is commonly used to test mice on a single time point. Therefore, we have a slightly more noisy measurement, especially for the first day of rotarod in this treatment schedule, which probably is the reason that we do not detect a significant difference here. This is now addressed in the related results section (lines 439-444)
Line 493 For non-specialists (of which I am one), a reference showing the inhibitory effect of fingolimod on SphK1 would be welcome.
We have now removed this statement. The literature on the effects of fingolimod on Sphk1 , Sphk2 and S1P is not conclusive in our view. We therefore now refer to fingolimod as an S1P modulator (as also indicated above).

Round 2
Reviewer 1 Report
Comments and Suggestions for Authors
The critiques have been adequately addressed.